

# Understanding mercury oxidation and air-snow exchange on the East Antarctic Plateau: A modeling study

Shaojie Song[1,*], Hélène Angot[2,3], Noelle E. Selin[1,2], Hubert Gallée[3], Francesca Sprovieri[4], Nicola Pirrone[5], Detlev Helmig[6], Joël Savarino[3], Olivier Magand[3], Aurélien Dommergue[3]

[1]Department of Earth, Atmospheric and Planetary Sciences, Massachusetts Institute of Technology, Cambridge, Massachusetts 02139, United States
[2]Institute for Data, Systems and Society, Massachusetts Institute of Technology, Cambridge, Massachusetts 02139, United States
[3]Univ. Grenoble Alpes, CNRS, IRD, Grenoble INP, Institut des Géosciences de l'Environnement (IGE), 38000 Grenoble, France
[4]CNR-Institute of Atmospheric Pollution Research, Division of Rende, Italy
[5]CNR-Institute of Atmospheric Pollution Research, Montelibretti, Rome, Italy
[6]Institute of Arctic and Alpine Research (INSTAAR), University of Colorado, Boulder, Colorado 80309-0450, USA
[*]Now at School of Engineering and Applied Sciences, Harvard University, Cambridge, Massachusetts 02138, United States

*Correspondence to*: Shaojie Song (songs@seas.harvard.edu)

**Abstract.** Distinct diurnal and seasonal variations of mercury (Hg) have been observed in near-surface air at Concordia station on the East Antarctic Plateau, but the processes controlling these characteristics are not well understood. Here, we use a box model to interpret the $Hg^0$ (gaseous elemental mercury) measurements in year 2013. The model includes atmospheric $Hg^0$ oxidation (by OH, $O_3$, or bromine), surface snow $Hg^{II}$ (oxidized mercury) reduction, and air-snow exchange, and is driven by meteorological fields from a regional climate model. The simulations suggest that a photochemically driven mercury diurnal cycle occurs at the air-snow interface in austral summer. The fast oxidation of $Hg^0$ in summer may be provided by a two-step bromine-initiated scheme, which is favored by low temperature and high nitrogen oxides at Concordia. The summertime diurnal variations of $Hg^0$ (peaking during daytime) may be confined within several tens of meters above the snow surface and affected by changing mixed layer depths. Snow reemission of $Hg^0$ is mainly driven by photoreduction of snow $Hg^{II}$ in summer. Intermittent warming events and a hypothesized reduction of $Hg^{II}$ occurring in snow in the dark may be important processes controlling the mercury variations in the non-summer period, although their relative importance are uncertain. The Br-initiated oxidation of $Hg^0$ is expected to be slower at Summit Greenland than at Concordia (due to their difference in temperature and levels of nitrogen oxides and ozone), which may contribute to the observed differences in the summertime diurnal variations of $Hg^0$ between these two polar inland stations.


## 1 Introduction

Mercury (Hg) is an environmental concern due to its health effects on humans and wildlife (Mergler et al., 2007). This trace element undergoes long-range transport in the atmosphere, and is readily cycled at the Earth's surfaces (Selin, 2009), and thus even the remote Antarctic plateau, a vast (about $5 \times 10^6 \, km^2$) and elevated (about 3 km above sea level) region of snow-covered
ice, receives significant mercury inputs (Dommergue et al., 2010).

Over the past decade, field studies have investigated mercury in air and/or snow at a few inland Antarctic stations, i.e., Concordia station (Dome C, 75°S 123°E), Dome Argus (80°S 77°E), Dome Fuji (77°S 40°E), and South Pole (90°S), as well as along several transects on the plateau (Brooks et al., 2008; Dommergue et al., 2012; Han et al., 2014; Li et al., 2014; Angot
et al., 2016b; Angot et al., 2016c; Wang et al., 2016; Han et al., 2017; Spolaor et al., 2018). Most of these studies only measured atmospheric mercury in austral summer, whereas Angot et al. (2016c) reported a year-round observational record at Dome C. All these measurements suggest that in summer (Nov–Feb), a photochemical mercury cycle occurs between the atmospheric boundary layer and surface snowpack, including the oxidation of gaseous elemental mercury ($Hg^0$) in air, the deposition of oxidized mercury ($Hg^{II}$) onto snow, the photoreduction of snow $Hg^{II}$, and the reemission of $Hg^0$ from the snowpack surface. A
clear diurnal cycle of $Hg^0$ (peaking at midday and decreasing to a minimum around midnight) was observed in near surface air, and has been attributed to enhanced $Hg^0$ reemission in the daytime as a result of increasing solar radiation (Dommergue et al., 2012; Angot et al., 2016c; Wang et al., 2016). The summertime photochemical mechanism of $Hg^0$ oxidation in air is unknown, but has been related to the high oxidizing capacity of the plateau, which is characterized by high concentrations of $NO_x$, OH, and other oxidants within the Antarctic mixed layers (Eisele et al., 2008; Helmig et al., 2008a; Helmig et al., 2008b;
Neff et al., 2008; Kukui et al., 2014; Frey et al., 2015). Interestingly, such summertime diurnal variations of $Hg^0$ have not been seen at the polar inland Summit Station atop the Greenland ice sheet (Brooks et al., 2011). As for other seasons, observations at Dome C showed high atmospheric $Hg^0$ in fall (Mar–Apr), exceeding those measured at the Antarctic coast and southern hemispheric mid-latitude sites. Such seasonal cycles were repeatedly measured in 2012–2015 at Dome C (Angot et al., 2016a). Moreover, in fall, the concentrations of $Hg^0$ peaked during the night. In winter (May–Aug), as expected, the diurnal cycle of
$Hg^0$ disappeared, and a gradual decline of $Hg^0$ was seen in near-surface air.

Overall, these observed seasonal and diurnal features of atmospheric mercury on the plateau are not well understood and not reproduced by global chemical transport models, likely due to their imperfect representations of boundary layer dynamics and chemical reaction pathways (Angot et al., 2016a). Here, we present detailed box model calculations to interpret observational
data collected at Dome C in 2013, and to explore important chemical and physical processes controlling diurnal and seasonal variations of atmospheric mercury. A better knowledge of these characteristics is helpful for evaluating the potential influence of the Antarctic plateau on the coastal environment (Bargagli, 2016), and for understanding processes occurring in other polar regions.



## 2 Methods

We have built a multiple-layer box model to account for mercury chemistry and transport in the lower troposphere and surface snow, and the exchange between them. Details on the model setup are given in this section. The modeling results are mainly compared with the measurement data of $Hg^0$ in year 2013. Briefly, $Hg^0$ concentrations were measured at three inlets (25, 210, and 1070 cm above surface) of a meteorological tower located in the "clean area" of Dome C (where snow is kept undisturbed). $Hg^0$ concentrations were also measured in the near-surface air and snow interstitial air with multi-inlet snow sampling manifolds (the so-called "snow towers"). The mercury measurements were performed using a Tekran 2537A automated analyzer (Tekran Inc., Toronto, Canada). The experimental details have been described in Angot et al. (2016c).

### 2.1 Model overview

The model accounts for vertical transport using outputs from a regional climate model (Sect. 2.2). As shown in Fig. 1, $Hg^0$ can be oxidized to $Hg^{II}$ by different gas-phase chemical schemes (Sect. 2.3). The photoreduction of $Hg^{II}$ in aqueous clouds and aerosols is not considered in the model because its mechanism is poorly understood, and also because the air above the plateau is cold and dry. The vertical resolution is ~2 m near the surface and gradually decreases with height above the surface, and there are 33 atmospheric layers in total below 500 m. In the free troposphere, $Hg^0$ and $Hg^{II}$ concentrations are prescribed (Sect. 2.4). $Hg^0$ and $Hg^{II}$ are transferred from air to snow through dry deposition (Sect. 2.5). Wet deposition is not considered due to low snow accumulation rates and large uncertainty in parameterizing this process (France et al., 2011; Palerme et al., 2017). The model tracks $Hg^0$ and $Hg^{II}$ in a surface snow reservoir, in which $Hg^{II}$ may be reduced to $Hg^0$ photolytically or in the dark (Sect. 2.5). The depth of the surface snow layer is set to 20 cm, equivalent to one to two $e$-folding light penetration depths at Dome C (France et al., 2011). The exchange of mercury between the surface snowpack and the deeper snowpack is not considered in the model because the photochemistry in the deeper snowpack is less active, and also because the diffusive transfer of $Hg^0$ between these two snow layers should be slower. Our model calculations are not expected to capture day-to-day variations since horizontal transport is ignored, and are thus compared with the average monthly and diurnal observations at Dome C as reported in Angot et al. (2016c). Different model scenarios are conducted by varying physiochemical processes and their parameters.

### 2.2 Meteorology

A surface-based temperature inversion layer exists at Dome C for most of the year, mainly due to radiation imbalance, while a convective mixed layer up to several hundred meters in depth develops during the daytime in summer in response to surface heating (see the Supplement, Sect. S1) (Pietroni et al., 2014). Here, the depth of the inversion/mixed layers is specified as ~500 m in our model, and the air above is regarded as the free troposphere. The vertical atmospheric transport is represented with turbulent diffusion coefficients ($K_z$) from the regional climate model MAR (Modèle Atmosphérique Régional) (Supplement, Sect. S1). The MAR data have been used to simulate several other atmospheric species (e.g., $NO_x$ and HONO) in the 2011–



2012 summer Oxidant Production in Antarctic Lands and Export (OPALE) campaign at Dome C (Legrand et al., 2014; Frey et al., 2015; Preunkert et al., 2015). In general, MAR simulations agree well with meteorological observations at Dome C (Gallée and Gorodetskaya, 2010; Gallée et al., 2015), whereas the intermittent warming events occurring primarily during the non-summer period, which decrease temperature inversion strength and strongly enhance vertical turbulence (leading to large $K_z$ values), may not be well represented. The vertical temperature gradients measured at a meteorological tower at Dome C indicate that the actual intensities of warming events should be weaker than results from MAR (Genthon et al., 2010). This is likely related to the cloud microphysical scheme in MAR, which is responsible for estimating the cloud cover and thus affects the estimation of surface temperature and buoyant forcing of turbulence. For example, in the wintertime, when the cloudiness is overestimated by the model, the downward infrared radiation is also overestimated. This overestimation limits surface cooling and subsequently the inhibition of turbulence, which is essentially generated by the wind shear. An accurate estimate of the warming events is challenging, and here we tentatively adjust MAR-modeled $K_z$ values during warming events using a rough empirical relationship between the temperature gradients and $K_z$, resulting in weaker exchange between the surface layers and free troposphere. It is important to note that such an adjustment is subject to large uncertainties and tends to underestimate the strength of vertical turbulence (Supplement, Sect. S1). Thus, owing to uncertainties in estimating warming events and their effects on the vertical transport of mercury in the non-summer period, both original and adjusted $K_z$ values are used to drive the mercury model in this study.

**2.3 Atmospheric mercury chemistry**

In the model, $Hg^0$ is oxidized in the atmosphere to $Hg^{II}$, while the oxidants, chemical kinetics, and oxidant concentrations are all uncertain. As shown in Table 1, the rate constants of $Hg^0$ reactions with $O_3$ (*R1*), OH (*R2*), and Br (*R3*) from existing theoretical and experimental studies may vary by factors of about 60, 8, and 4, respectively. While used in several chemical transport models, $O_3$ and OH based chemical mechanisms are unlikely as pure gas phase reactions since the formation of HgO is endothermic (Subir et al., 2011). The two-step Br-initiated scheme (*R3–R10*) can explain polar atmospheric mercury depletion events (Sprovieri et al., 2005; Steffen et al., 2008), and is likely the dominant $Hg^0$ oxidation pathway globally (Holmes et al., 2006; Horowitz et al., 2017; Ye et al., 2018). The recombination of $Hg^0$ and Br forms unstable $Hg^I$Br, which either dissociates or is oxidized to $Hg^{II}$ by $NO_2$, $HO_2$, OH, Br, or BrO. The effective oxidation rate constant of this two-step scheme is expressed in Eq. (1), assuming a steady state of $Hg^I$Br, as it forms slowly by *R3*, and is oxidized readily by *R6–R10*, where terms in brackets refer to concentrations, and $k_3$–$k_{10}$ are reaction rates of *R3–R10*. The gas phase oxidations of $Hg^0$ by other species and the aqueous and heterogeneous processes are not considered here (Supplement, Sect. S2) (Lin and Pehkonen, 1999; Subir et al., 2011; Ariya et al., 2015).

$$k_{\text{eff}} = \frac{k_3[\text{Br}] \cdot (k_6[\text{NO}_2] + k_7[\text{OH}] + k_8[\text{HO}_2] + k_9[\text{Br}] + k_{10}[\text{BrO}])}{k_4 + k_5[\text{Br}] + k_6[\text{NO}_2] + k_7[\text{OH}] + k_8[\text{HO}_2] + k_9[\text{Br}] + k_{10}[\text{BrO}]} \quad (1)$$



Concentrations of chemical species, including $O_3$, $HO_x$ (OH, $HO_2$), $BrO_x$ (Br, BrO), and $NO_x$ (NO, $NO_2$), are prescribed based on the available measurements and global chemical transport model (CTM) simulations (details in the Supplement, Sect. S3). Monthly and diurnal averages are computed. $O_3$ and $NO_x$ are specified based on measurements in near-surface air (Angot et al., 2016c; Legrand et al., 2016a; Helmig et al., 2018), and a uniform $O_3$ vertical profile is assumed, consistent with aircraft observations on the plateau (Slusher et al., 2010; Legrand et al., 2016a). The $NO_x$ vertical profile has not been measured and is estimated assuming an exponential decay with height starting at the surface (Slusher et al., 2010). The previously reported potential bias in the measurement ratios of [NO]/[$NO_2$] (Frey et al., 2015) does not significantly affect our model results, as suggested by a sensitivity test. The $HO_x$ concentrations in summer are set based on measurements from the OPALE campaign, and their values in other seasons are estimated using relationships with $J(NO_2)$ and NO (Kukui et al., 2014). The uncertainties in $O_3$ and OH concentrations are assumed to be 2% and 50%, respectively, as inferred from *in situ* measurements at Dome C (Kukui et al., 2014).

For BrO concentrations, due to lack of measurements, we rely on two global CTMs, GEOS-Chem and *p*-TOMCAT (Yang et al., 2005; Sherwen et al., 2016). We assume no diurnal and vertical variations of BrO (Stutz et al., 2011; Legrand et al., 2016b). The modeled BrO mixing ratios from these two CTMs are similar, less than 0.1 pptv in winter and ~0.4 pptv in other seasons (Supplement, Fig. S8). The modeled BrO is likely at the lower limits of its uncertainty range, as suggested by the comparison of the modeled tropospheric BrO columns and their values retrieved from the GOME-2 satellite (Sherwen et al., 2016). Legrand et al. (2016b) measured total inorganic gaseous bromine concentrations at Dome C and suggested that the upper limit of BrO is ~1 pptv. Based on the above information, the uncertainty of BrO concentrations is set as a factor of 2.5. It is important to note that the seasonal patterns of the modeled BrO by the CTMs may have biases, as indicated by the total inorganic bromine measurements at Dome C (Legrand et al., 2016b). The modeled BrO is likely biased high in fall and spring, which affects $Hg^0$ concentrations simulated by the mercury model (Sect. 3.4). The concentrations of Br are estimated assuming a photochemical steady state: $[Br]/[BrO]=(J_{BrO}+k_{BrO+NO}[NO])/(k_{Br+O3}[O_3])$ (Holmes et al., 2010), where $J_{BrO}$ is the BrO photolysis frequency, and $k_{BrO+NO}$ and $k_{Br+O3}$ are rate constants for BrO + NO → Br + $NO_2$, and Br + $O_3$ → BrO + $O_2$, respectively (Sander, 2011).

## 2.4 Mercury concentrations in the free troposphere

Due to lack of measurements, we rely on two global CTMs, GEOS-Chem (version 9-02) and GLEMOS, to specify the free tropospheric mercury concentrations (Angot et al., 2016a; Travnikov et al., 2017). The former uses a Br oxidation scheme, whereas the latter assumes OH and $O_3$ to be the oxidants of $Hg^0$. Monthly $Hg^0$ and $Hg^{II}$ concentrations at 500 m above ground level in the Dome C grid box are extracted from these two CTMs. Studies have identified that the CTMs show significant seasonal biases in modeled mercury concentrations, when compared to mercury observations at two southern hemispheric background stations, Amsterdam Island (38°S 78°E) and Cape Point (34°S 18°E) (Angot et al., 2014; Song et al., 2015; Horowitz et al., 2017; Martin et al., 2017), implying potential biases in modeled mercury budgets for the southern hemisphere. Hence, we adjust the modeled free tropospheric mercury concentrations using the scaling factors estimated by model-



observation comparisons for these two background stations: $R_{i,j} = \overline{X_{obs,i,j}}/\overline{X_{mod,i,j}}$, where $\overline{X}$ represents the average mercury concentrations, and $i$ and $j$ indicate each month and model, respectively. The two CTMs predict similar total gaseous mercury ($Hg^T = Hg^0 + Hg^{II}$) concentrations with annual means of ~1.0 ng m$^{-3}$, whereas the modeled $Hg^{II}$ concentrations during the sunlit period are much higher in GEOS-Chem than in GLEMOS due to their different chemical mechanisms (Supplement, Fig.

S9). In our simulations, the free tropospheric mercury data are chosen from either GEOS-Chem or GLEMOS according to the chemical oxidation scheme ($O_3$, OH, or Br) used in each model scenario, for consistency. For example, the GEOS-Chem free tropospheric mercury data are used when the Br scheme is assumed in the box model simulation. Both CTMs use reaction rate constants at the lower limits. When the upper-limit reaction rates are assumed in the model scenarios, we expect more mercury should exist in its oxidized form, $Hg^{II}$, in the free troposphere, and thus, we adjust free tropospheric concentrations of $Hg^0$ and

$Hg^{II}$ according to this equation: $Hg^{II}_{upper}/Hg^0_{upper} = R \times \left(Hg^{II}_{lower}/Hg^0_{lower}\right)$, where $R$ is the ratio between the upper- and lower-limit reaction rate constants, whereas the total $Hg^T$ concentrations remain unchanged.

**2.5 Air-snow mercury exchange and snow mercury transformation**

Dry deposition fluxes of $Hg^0$ and $Hg^{II}$ are determined by their concentrations at the atmospheric ground level and prescribed deposition velocities ($V_d$). The effects of wind speeds and snow properties on $V_d$ are not included here due to lack of

information. As indicated by previous studies (Lindberg et al., 2002; Brooks et al., 2006; Skov et al., 2006), the values of $V_d$ for $Hg^0$ and $Hg^{II}$ are set to $1 \times 10^{-4}$ and 1 cm s$^{-1}$, respectively (Zhang et al., 2009). These $V_d$ parameters are not well constrained, but we find that varying the values of $V_d$ by a factor of 2 does not change the main findings of this study. For $Hg^0$, the bidirectional fluxes between surface snow and air are considered and estimated by $Hg^0$ concentration differences and the turbulent and molecular diffusion coefficients in the snow interstitial air. Following Durnford et al. (2012), the molecular

diffusion coefficient ($D_m$) in our model is set to $6 \times 10^{-6}$ m$^2$ s$^{-1}$. The turbulent diffusion coefficients ($D_t$) can be estimated by an explicit representation of the vertical wind pumping within the snowpack, which include several uncertain parameters, such as the height and wavelength of sastrugi (snow-eroded grooves or ridges) and the permeability of surface snowpack (Cunningham and Waddington, 1993; Thomas et al., 2011; Zatko et al., 2013; Toyota et al., 2014b). The estimated values of $D_t$ using this approach and the air and snow properties at Dome C may vary from the order of $10^{-6}$ to $10^{-4}$ m$^2$ s$^{-1}$ for the surface

snowpack with a depth of 20 cm. Here, a more simple approach is adopted following Durnford et al. (2012), in which $D_t$ is set proportional to the atmospheric ground level turbulent kinetic energy (TKE) obtained from the MAR model: $D_t = $ TKE (m$^2$ s$^{-2}$) $\times 3 \times 10^{-3}$ s. $D_t$ varies by season and by time of day and has an annual median value of $3 \times 10^{-4}$ m$^2$ s$^{-1}$. The choice of the scaling factors ($3 \times 10^{-3}$ s by default in the model) is found to influence the modeled $Hg^0$ concentrations in the snow interstitial air (Sect. 3.2).


Previous studies have suggested that $Hg^{II}$ can be reduced both photolytically and in the dark, and the photolytic and dark oxidation of $Hg^0$ may also occur, but the reaction rates and reductants/oxidants of individual pathways are largely unknown



(for a review, see Durnford and Dastoor (2011)). Sunlight, in particular UV-B (280–320 nm) radiation, greatly enhances the formation of $Hg^0$ (Poulain et al., 2004; Dommergue et al., 2007; Johnson et al., 2008). Similar to previous models (Durnford et al., 2012; Toyota et al., 2014a), we include a first-order photoreduction of $Hg^{II}$ in the surface snowpack and scale its rate by $J(O(^1D))$, the photolysis frequency of $O_3$. In doing so, we assume that the supply of reductants is ample and that all $Hg^{II}$ is reducible (Durnford and Dastoor, 2011). The photoreduction rate is poorly constrained, with a corresponding lifetime (denoted as $\tau_{PR}$) from a few days to several weeks (Toyota et al., 2014a). We also include dark reduction of snow $Hg^{II}$ (the corresponding lifetime denoted as $\tau_{DR}$) in our model simulations for the non-summer period (Sect. 3.4).

## 3 Results and discussion

### 3.1 Atmospheric $Hg^0$ oxidation rates

We have computed ranges of atmospheric $Hg^0$ oxidation rates for different schemes ($O_3$, OH, and two-step Br), using the low (i.e., lower limit) and high (i.e., upper limit) rate constants listed in Table 1 and uncertainties of oxidant concentrations (Sect. 2.3). As shown in Fig. 2, the $Hg^0$ oxidation rates for these schemes in the inversion/mixed layers have large uncertainty ranges. Since the OH and Br concentrations are largely determined by the amount of solar radiation, the oxidation rates of Hg under these schemes exhibit strong seasonal and diurnal variations, while the $O_3$ scheme does not. In austral summer (Nov–Feb), the two-step Br oxidation scheme (corresponding $Hg^0$ oxidation lifetimes denoted as $\tau_{OX} \sim 1.7–22$ days) is more efficient than the $O_3$ ($\tau_{OX} \sim 19–1300$ days) and OH ($\tau_{OX} \sim 17–350$ days) oxidation schemes. We find that the fast two-step Br oxidation is favored by low ambient temperature, high concentrations of $NO_x$, and low concentrations of $O_3$ at Dome C. This is because the thermal dissociation rates of the intermediate $Hg^IBr$ decrease rapidly at a lower temperature, and because the concentrations of Br are influenced by the concentrations of $NO_x$ and $O_3$ (Sect. 2.3). In austral winter (May–Aug), by contrast, the $O_3$ oxidation scheme ($\tau_{OX} \sim 13–900$ days) is usually more efficient than the others. A series of combinations of oxidation schemes, oxidant concentrations, and chemical kinetics are tested in our model simulations.

### 3.2 Strong photochemistry in summer

During the summer months, the observed $Hg^0$ concentrations in near-surface Dome C air show a pronounced diurnal pattern, which usually peaks in the daytime and minimizes at night, as shown in Fig. 3 and Fig. S10 in the Supplement. The amplitudes of diurnal variations of observed $Hg^0$ reach ~0.4 ng m$^{-3}$ in January and ~0.3 ng m$^{-3}$ in February and November, respectively, higher than other seasons. This characteristic has been attributed to enhanced reemissions of $Hg^0$ in the daytime (Angot et al., 2016c; Wang et al., 2016), highlighting a dynamic Antarctic surface snowpack. The solar zenith angle has a diurnal cycle during summer, and a convective layer develops in the daytime as a response to surface heating, enhancing strengths of vertical mixing and snow ventilation. Previous studies have suggested rapid recurring cycles of oxidation and reemission of $Hg^0$ in summer, but chemical mechanisms have not been well defined (Angot et al., 2016c; Wang et al., 2016). As photochemical processes in the air and surface snow are of obvious importance for summer, we have conducted a series of mercury model





sensitivity simulations by varying atmospheric oxidants (O$_3$, OH, or Br), their concentrations (high or low) and chemical reaction rate constants (upper or lower), and surface snow Hg$^{II}$ photoreduction rates ($\tau_{PR}$ from three days to three weeks). In total, we ran 24 model sensitivity scenarios. Through comparing modeling results to observations, key atmospheric Hg$^0$ oxidants may be identified, and surface snow Hg$^{II}$ photoreduction rates may be constrained. Some of these scenarios have

large biases compared to observations for the non-summer months, which is likely due to several factors in these simulations that will be discussed in detail in Sect. 3.4: (1) the adjusted $K_z$ values during the warming events are used, which tends to underestimate the mercury vertical transport from the free troposphere, (2) the Br concentrations used in the model calculations are likely overestimated in the non-summer period, and/or (3) the dark reduction of snow Hg$^{II}$, which may be important in the non-summer period, is not included.

The modeled Hg$^0$ concentrations in near-surface air from various scenarios are compared to observations in Fig. 3 and in the Supplement, Sect. S4 (only the data collected at 25 cm above surface are shown, and the model-observation comparison results for the data at 210 and 1070 cm are similar). We find, during summer, that model scenarios using either OH or O$_3$ oxidation schemes do not reproduce the diurnal variations of Hg$^0$, and tend to overestimate atmospheric Hg$^0$ concentrations, even when

high oxidant concentrations and upper-limit reaction rates are assumed (resulting in $\tau_{OX}$ ~ 20 days). Among the scenarios with the bromine oxidation scheme, BR_HH_14d (using high Br concentrations and upper-limit reaction rate constants; $\tau_{OX}$ ~ 2 days and $\tau_{PR}$ of 2 weeks in summer) best reproduces the concentrations of atmospheric Hg$^0$ and its diurnal patterns during the summer months (calculated normalized root-mean-square errors of < 20%; Supplement, Sect. S4). This scenario shows larger Hg$^0$ diurnal variations in Jan–Dec than Feb–Nov, consistent with observations (Angot et al., 2016c; Spolaor et al., 2018). The

differences in solar radiation in these summer months are expected to influence the strength of photochemical activities (such as Br concentration and photoreduction rates of snow Hg$^{II}$). Therefore, these sensitivity simulations suggest that a fast oxidation for atmospheric Hg$^0$ occurs in the surface layers at Dome C in summer, and that the fast oxidation of Hg$^0$ may be provided by a two-step Br scheme with its upper-limit reaction rates.

The summertime average Hg$^0$ concentrations modeled by the scenario BR_HH_14d are also compared with those measured at different sampling heights, as shown in Fig. 4. The snow tower measurements indicate that Hg$^0$ concentrations in the surface snow interstitial air (10 cm below surface) are about 0.2 ng m$^{-3}$ higher than those in the air (50 cm above surface). The model predicts a similar Hg$^0$ difference of about 0.3 ng m$^{-3}$. These results suggest the snow-to-air transport of Hg$^0$ and the production of Hg$^0$ in the surface snowpack. It is noted that the modeled difference in Hg$^0$ concentrations depends on the assumed turbulent

diffusion coefficients ($D_t$). Larger $D_t$ implies faster vertical mixing of Hg$^0$, and thus corresponds to smaller differences between the surface snowpack and atmosphere (Supplement, Fig. S12). The measured Hg$^0$ concentrations in the interstitial air of the deeper snowpack are lower than those in the surface snowpack, suggesting that the production of Hg$^0$ may mainly occur in the snow near surface. In the model, the production of Hg$^0$ in surface snow arises from the photoreduction of Hg$^{II}$ during summer. The photoreduction rates of surface snow Hg$^{II}$ in BR_HH_14d ($\tau_{PR}$ of 2 weeks) agree well with observations at South Pole in



Brooks et al. (2008), who estimated a lifetime of surface snow mercury of ~16 days. The surface snow mercury concentrations modeled by BR_HH_14d are ~20 ng L$^{-1}$ (Supplement, Fig. S13). The available measurements suggest that surface snow mercury concentrations were highly variable, ranging from ~ 3 to 50 ng L$^{-1}$ (Angot et al., 2016c; Spolaor et al., 2018).

5    The summertime vertical and diurnal profiles of modeled Hg$^0$ concentrations in near-surface air are shown in Fig. 5a. Model results are from the scenario BR_HH_14d (using high Br concentrations and upper-limit reaction rates; $\tau_{OX}$ ~ 2 days and $\tau_{PR}$ of 2 weeks), which best reproduces the observed Hg$^0$ in summer. We find that the diurnal variation ranges of Hg$^0$ are greater than 0.2 ng m$^{-3}$ only for near-surface levels from snow to about 50 m above. As shown in Fig. 5b, the summertime Hg$^0$ cycles in the inversion/mixed layers are primarily driven by diffusion from snow and oxidation loss. The dry deposition and transport from the free troposphere are insignificant. The amplitude of Hg$^0$ oxidation loss increases during the daytime due to enhanced photochemical activities. Diffusion of Hg$^0$ from surface snow is controlled by the rate of snow Hg$^{II}$ photoreduction, which also peaks in the daytime. The diurnal profiles of the modeled Hg$^0$ fluxes from simulations using the O$_3$ and OH oxidation schemes are given in the Supplement, Fig. S14. As expected, the amplitudes of their fluxes are much smaller than this bromine oxidation model scenario. In order to elucidate the drivers of strong diurnal variations of Hg$^0$ in near-surface vertical levels in summer, 15    we calculated the diurnal cycles of Hg$^0$ concentrations and all related fluxes for 0–50 meters above snow (Figs. 5c and 5d). The net diffusion of Hg$^0$ refers to difference in its diffusion from snow and to upper levels. The latter is controlled by the changing mixed layer heights, which are low at night (< 50 m) and strongly increased during the daytime (Angot et al., 2016c). Thus, at night, all Hg$^0$ diffused from snow remains inside the shallow mixed layer, while in the daytime a large fraction is transferred to the air above 50 m. The net Hg$^0$ flux, the derivative of its diurnal variation, is determined by the net diffusion and oxidation loss of Hg$^0$. As shown in Fig. 5d, the net flux is positive in the morning, but becomes negative in the afternoon, thus leading to the Hg$^0$ maximum around noon. Overall, the diurnal variations of Hg$^0$ in near surface levels in summer are determined by the changes in the Hg$^0$ oxidation loss, snow Hg$^{II}$ photoreduction, and mixed layer depth, all of which are in turn controlled by the strong photochemical activity during this time period at Dome C.

25    Furthermore, our model results suggest that the air above Dome C is enriched in Hg$^{II}$ during summer, consistent with its strong photochemical activity. As shown in Fig. 6, the predicted Hg$^{II}$ by the scenario BR_HH_14d increases with height, from ~0.1 ng m$^{-3}$ near surface to ~0.5 ng m$^{-3}$ at 500 m. Such Hg$^{II}$ concentrations are comparable to the levels identified in the upper free troposphere for the mid-latitudes (Bieser et al., 2017). A diurnal pattern of Hg$^{II}$ with higher concentrations in the afternoon is predicted in near-surface air by the model. These characteristics should be verified by future measurement studies. Preliminary 30    filed sampling using polyethersulfone cation-exchange membranes in a 2014/2015 summer campaign obtained Hg$^{II}$ of about 0.4 ng m$^{-3}$ (average concentration from 3 filter samples) (Angot, 2016).



### 3.3 Comparison with summertime data at Summit, Greenland

Dome C (75°S 123°E, 3 km above sea level) and Summit Greenland (73°N 38°W, 3.2 km above sea level) are both located in high altitude and far from the ocean (hundreds of kilometers). As a result, their meteorological and chemical conditions have similarities. In summer, both stations have shallow boundary layers that are stable at night but convective during the day

(Helmig et al., 2002; Cohen et al., 2007; Van Dam et al., 2013). Active bromine chemistry was found to occur at Summit in summer (Thomas et al., 2011), and the average BrO mixing ratios in near-surface air were 0.9–1.5 pptv (Liao et al., 2011; Stutz et al., 2011), comparable to the 1 pptv upper limit at Dome C (Legrand et al., 2016b). Thus, it is expected that these two stations may have similar mercury variabilities in near-surface air. Brooks et al. (2011) measured atmospheric mercury concentrations in 2007–2008 summer at Summit, but did not observe a significant diurnal cycle of $Hg^0$ peaking at noon as was

seen at Dome C. Based on our model analysis, we can identify several potential factors that can contribute to differences in the diurnal cycles of $Hg^0$ between these two inland polar locations.

First, although BrO concentrations at Summit are comparable or higher than at Dome C, the concentrations of Br at Summit, the primary oxidant of $Hg^0$, may be much lower. As described in Sect. 2.3, the [Br]/[BrO] ratios are positively related to the

concentrations of NO and negatively related to the concentrations of $O_3$. Reported summertime $NO_x$ concentrations at Summit (~20 pptv) are lower than at Dome C (~300 pptv), whereas $O_3$ at Summit (~50 ppbv) is approximately two times that at Dome C (~25 ppbv) (Helmig et al., 2008a; Frey et al., 2015; Kramer et al., 2015; Van Dam et al., 2015; Huang et al., 2017). The larger $NO_x$ concentrations at Dome C have been suggested to arise in part from larger $NO_x$ emissions from surface snow, which are in turn driven by the photolysis of nitrate in the surface snowpack (Frey et al., 2015). A back-of-the-envelope calculation

shows, assuming the same BrO concentrations, that Br concentrations at Dome C would be on average a factor of 6 higher than at Summit. Second, the thermal dissociation rate of the intermediate $Hg^I Br$ at Summit should be one order of magnitude greater than that at Dome C. This is because this rate strongly depends on temperature (Table 1), and the ambient temperature at Summit is about 15 K higher than at Dome C. Third, the oxidation of $Hg^I Br$ by $NO_2$ (the dominant second step oxidant) is significantly slower at Summit than at Dome C, due to their different concentrations of $NO_2$. In fact, the rates of oxidation by

$NO_2$ and dissociation of $Hg^I Br$ are comparable at Summit. This is in contrast with Dome C, where the oxidation by $NO_2$ can easily outcompete the thermal dissociation of $Hg^I Br$. All in all, we expect that the Br-initiated oxidation of $Hg^0$ should be slower at Summit than at Dome C, leading to weaker oxidation/reemission cycling of $Hg^0$ during summer. It is also noted that atmospheric circulation on Greenland may be influenced by stronger synoptic scale events than over the Antarctic plateau, because the air is thicker over the Greenland ice sheet (leading to a weaker decrease of relative vorticity when a large scale

eddy propagates from the ice sheet margin towards the center). However, the impact of this circulation pattern on the diurnal cycle of $Hg^0$ is unclear.





## 3.4 Non-summer period

We showed above that the model simulations including the photoreduction of snow $Hg^{II}$ and a fast bromine oxidation of atmospheric $Hg^0$, could reasonably explain the observed atmospheric mercury variations during summer. However, these simulations strongly underestimate $Hg^0$ concentrations in the non-summer months (Fig. 3), when solar radiation is weakened or completely absent. Based on our understanding of air and snow mercury cycling (Fig. 1), such model-observation discrepancies may imply, for the non-summer period, that in the model the vertical transport of mercury from the free troposphere is underestimated, the reduction of snow $Hg^{II}$ is underestimated, and/or the oxidation of atmospheric $Hg^0$ is overestimated. All these processes are poorly constrained in the non-summer period in part because previous studies have mainly focused on the summer season. The model performance can be improved by modifying the representation of these processes.

First of all, it is important to note in the above simulations that the adjusted $K_z$ values in the warming events are used to drive the mercury model, which tends to underestimate the transport of mercury from the free troposphere. We therefore conducted a sensitivity simulation (BR_S1) to examine the possible effects of warming events on modeling results. The difference between BR_S1 and BR_HH_14d (using high Br concentrations and upper-limit reaction rates; $\tau_{OX}$ ~ 2 days and $\tau_{PR}$ of 2 weeks in summer) is that the original MAR-modeled $K_z$ values are used in BR_S1, which may overestimate the transport of mercury form the free troposphere. As shown in Fig. 7a, in the non-summer months, near-surface air $Hg^0$ concentrations by BR_S1 are close to the prescribed $Hg^0$ concentrations in the free troposphere, and are significantly higher than those from BR_HH_14d. However, the scenario BR_S1 cannot reproduce the high atmospheric $Hg^0$ concentrations of ~1.2 ng m$^{-3}$ in fall (exceeding its levels at the Antarctic coastal regions and the southern hemispheric mid-latitude sites) and the diurnal cycles of $Hg^0$ in fall peaking in the night. This result indicates that $Hg^0$ may be produced below the atmospheric mixed layers at Dome C. In addition, surface snow Hg concentrations by BR_S1 exhibit an increase during the non-summer period (Fig. 7b), as a result of $Hg^{II}$ transport in warming events from the free troposphere (Fig. 7c). The deposited $Hg^{II}$ is accumulated in surface snow (photoreduction of $Hg^{II}$ is weak in the non-summer period). Such an enhancement of snow mercury was not measured at Dome C (Angot et al., 2016c). Therefore, we postulate that the existence of warming events during the non-summer period can significantly enhance $Hg^0$ concentrations in near-surface air, but is unlikely to be the only reason for the observed mercury variations.

Second, the reduction of snow $Hg^{II}$ might occur in the dark, which would produce $Hg^0$ and sustain atmospheric concentrations of $Hg^0$ through snow-to-air diffusion and convective transport. The possibility of the presence of dark reduction has been reported in previous laboratory and field studies (Lalonde et al., 2003; Ferrari et al., 2004; Dommergue et al., 2007; Faïn et al., 2007), although actual mechanisms remain unclear. The reduction might be a continuation of photolytically initiated reactions or through reactions requiring no insolation at all (Durnford and Dastoor, 2011). The HO$_2$ radical produced in the dark surface





snowpack may serve as a potential Hg$^{II}$ reductant (Dommergue et al., 2003; Ferrari et al., 2004). The dark reduction rates estimated in these studies are much lower than the photoreduction rates of Hg$^{II}$. Some observational evidence at Dome C supports the hypothesis of snow Hg$^{II}$ dark reduction. Near-surface air Hg$^0$ concentrations peaked in the night in fall, and Hg$^0$ concentrations in snow interstitial air were higher than air Hg$^0$ in fall and winter (Angot et al., 2016c). Thus, we have conducted

a sensitivity simulation, BR_S2, which added a first-order dark reduction of snow Hg$^{II}$ based on BR_HH_14d, in order to examine the possible effects of dark reduction on model results. The reaction rate corresponds to an average $\tau_{DR}$ of ~1 year for the non-summer period, and is scaled by NO$_x$ concentrations since this process is likely related to nitrogen chemistry. As shown in Fig. 8, the hypothesized snow Hg$^{II}$ dark reduction process leads to a small increase in the snow-to-air diffusive fluxes of Hg$^0$ ($< 0.5$ ng m$^{-2}$ h$^{-1}$), which can increase the concentrations of atmospheric Hg$^0$ in the non-summer period, especially in

winter. This scenario also better reproduces the diurnal variation of Hg$^0$ in the fall months.

Third, oxidation of atmospheric Hg$^0$ may be overestimated in our model in the non-summer period. As described in Sect. 2.3, the modeled BrO concentrations by the CTMs may have seasonal biases. Total inorganic bromine measurements at Dome C (Legrand et al., 2016b) suggested that the modeled BrO is likely biased high by up to a factor of 3 in fall and spring. The

reasons remain unknown, but are probably related to several factors, including depositions of Br-containing species, snow reemission or long-distance transport of Br$_2$/BrCl, and photochemical Br reactions (Yang X., British Antarctic Survey, personal communication, Jan 2017). A sensitivity simulation shows that reducing BrO concentrations in fall could increase the modeled air Hg$^0$ concentrations during the fall and winter months (Fig. S15 in the Supplement).

Based on the above sensitivity analysis, we find that the all these three processes (intermittent warming events, dark reduction of snow mercury, and overestimation of bromine oxidation) can help explain the observed high mercury concentrations in the non-summer period. Their relative contributions, however, are difficult to constrain since the understanding of these processes is limited.

## 4 Summary and future research needs

We have conducted box model calculations to explore important chemical and physical processes controlling the diurnal and seasonal variations of mercury at Dome C. The atmospheric Hg$^0$ oxidation rates of the OH, O$_3$, and the two-step Br-initiated schemes all have large uncertainty ranges due to uncertain chemical kinetics and oxidants concentrations. In austral summer, the Br oxidation scheme, favored by low ambient temperature and high concentrations of NO$_x$, is more efficient than the OH and O$_3$ schemes. The model simulations support the hypothesis that rapid recurring cycles of oxidation and reemission of Hg$^0$

occur in summer. Among the model scenarios tested, the simulations using the Br oxidation scheme (with upper-limit reaction rates) can best match mercury observations in summer. The modeling results indicate that strong diurnal variations of Hg$^0$ in summer may be confined within several tens of meters above the snow surface, and are primarily determined by changes in



Hg$^0$ oxidation loss, snow Hg$^{II}$ photoreduction, and mixed layer depths. For the non-summer period, the model-observation comparisons at Dome C suggest the intermittent warming events and a hypothesized dark reduction of snow Hg$^{II}$ may be important processes controlling the mercury variations, but their relative importance is uncertain. The Br-initiated oxidation of Hg$^0$ is expected to be slower at Summit Greenland because of high temperatures, high O$_3$, and low NO$_x$ conditions, which might contribute to the observed differences in the summertime diurnal variations of Hg$^0$ between these two polar inland locations.

In order to obtain a better understanding of mercury cycling over the East Antarctic plateau, we suggest several areas for future research. It is essential to better constrain the concentration levels of bromine species, especially BrO$_x$, through more field experiments and modeling studies. It is also important to reduce uncertainties in existing chemical kinetic parameters of bromine oxidation mechanisms. Our modeling indicates relatively high atmospheric Hg$^{II}$ concentrations in summer, which remains to be verified by field measurements. A better characterization of atmospheric vertical transport during the non-summer period is needed. The chemical mechanisms for snow mercury processes, including photo- and dark-reduction, should also be further investigated. Given the rapid exchange of mercury between the surface snowpack and the atmosphere (especially during summer), regional modeling studies should be conducted in the future in order to understand the total and speciated mercury budgets over the entire Antarctic plateau and the influence of the plateau on the coastal environments.

**Data availability**

The mercury box model is available at *http://github.com/shaojiesong/Hg_DomeConcordia*. The mercury measurement data at Dome C are available upon request at *http://sdi.iia.cnr.it/geoint/publicpage/GMOS/gmos_historical.zul*. The ozone and NO$_x$ measurement data at Dome C are available upon request to the authors.

**Competing interests**

The authors declare that they have no conflict of interest.

**Acknowledgments**

The authors acknowledge the MISTI Global Seed Funds (MIT-France). The model work at MIT is supported by the U.S. NSF Atmospheric Chemistry Program #1053648. Mercury measurements at Concordia station were initiated with the FP7 GMOS project and are supported by IPEV GMOstral Program 1028. D. H. acknowledges logistical and financial support by the French Polar Institute (Program 1011, SUNITEDC), and support from the U.S. NSF Office of Polar Programs through grant #1142145. A.D. and J.S. thank LEFE/INSU for their financial support. S.S thanks the MIT Henry Houghton Fund, and H.A. the Univ. Grenoble Alpes Doctoral School for Earth, Planetary and Environmental Sciences for travel support. We thank Tomás Sherwen




for providing GEOS-Chem bromine model results, Xin Yang for *p*-TOMCAT bromine model results, Oleg Travnikov for GLEMOS mercury model results, and Jack Dibb for access to the measurement data from Summit. We thank Susan Solomon, Ronald Prinn, Daniel Jaffe, Jennie Thomas, Warren Cairns, Jeroen Sonke, Roberto Grilli, Michel Legrand, and Christopher Holmes for comments and/or helpful discussions.

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



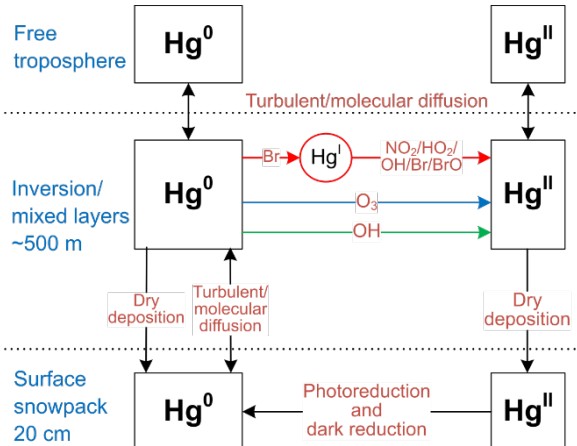

**Figure 1. Chemical and physical processes represented in the mercury box model.** $Hg^0$ can be oxidized to $Hg^{II}$ by three different gas-phase chemical schemes (OH, $O_3$, or a two-step Br-initiated scheme). Note that the concentrations of the intermediate $Hg^I$ in the two-step Br-initiated oxidation mechanism are not tracked since its lifetime is short, and thus effective reaction rates are used to describe the oxidation of $Hg^0$ to $Hg^{II}$ for this mechanism (Sect. 2.3).

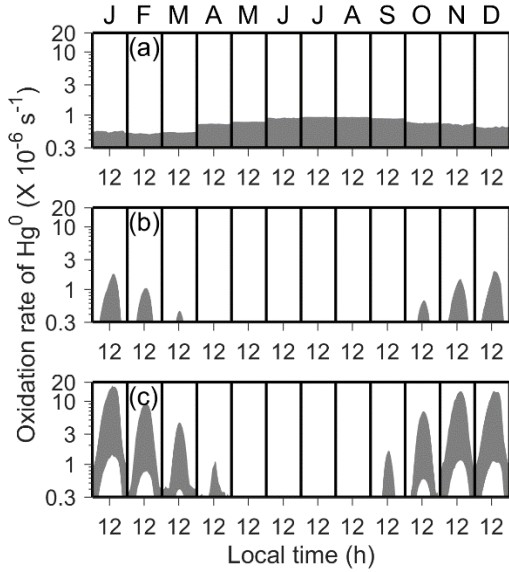

**Figure 2. Uncertainty ranges of atmospheric $Hg^0$ oxidation rates within the inversion/mixed layers.** (a) $O_3$, (b) OH, and (c) Br. Monthly and diurnal variations in year 2013 are shown in the shaded regions. Note that the *y* axis is in log scale.





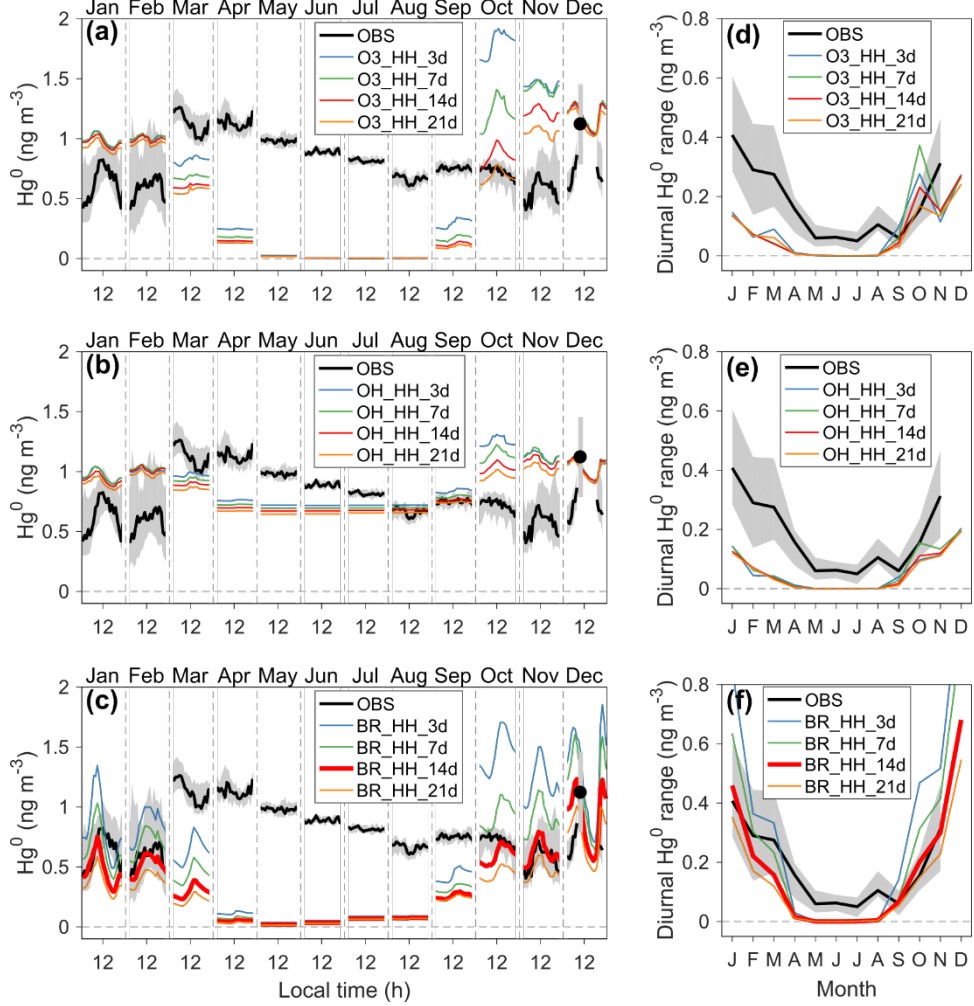

**Figure 3. Comparison of seasonal and diurnal variations of near-surface atmospheric Hg⁰ concentrations between observations and model.** (a–c) show monthly and diurnal Hg⁰ observations in year 2013 and modeling results from different scenarios. (d–f) show diurnal Hg⁰ ranges calculated from the maximum and minimum hourly concentrations in each month. The shaded regions indicate 25% and 75% percentiles in observations. Observations were conducted at 25 cm above snow surface at Dome C. The name of each scenario reflects the atmospheric oxidant, its concentration levels, chemical reaction rates (H = *high* or *upper*, L = *low* or *lower*), and the photoreduction rates of snow mercury (in days). For example, the scenario with name "O3_HH_3d" assumes O₃ as the oxidant, and high oxidant concentrations and high reaction rates are applied, and $\tau_{PR}$ is set to three days.





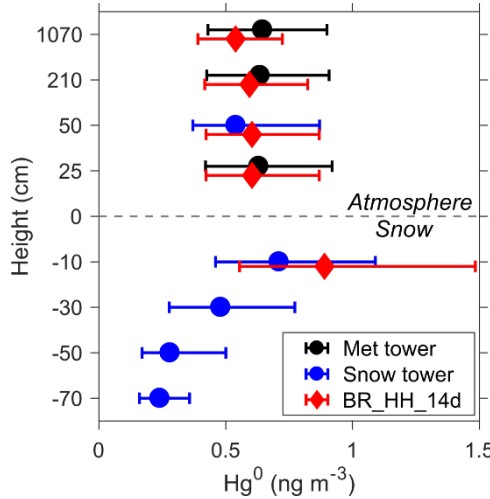

**Figure 4. Summertime average Hg⁰ concentrations at different heights from observations and model.** The observations include the meteorological tower (25, 210, and 1070 cm above snow surface) and snow tower (50 cm above snow surface and 10, 30, 50, and 70 cm below snow surface). Model results from the scenario BR_HH_14d are shown. Measurement data are from the snow tower #1 as reported in Angot et al. (2016c). Error bars indicate 25% and 75% percentiles.

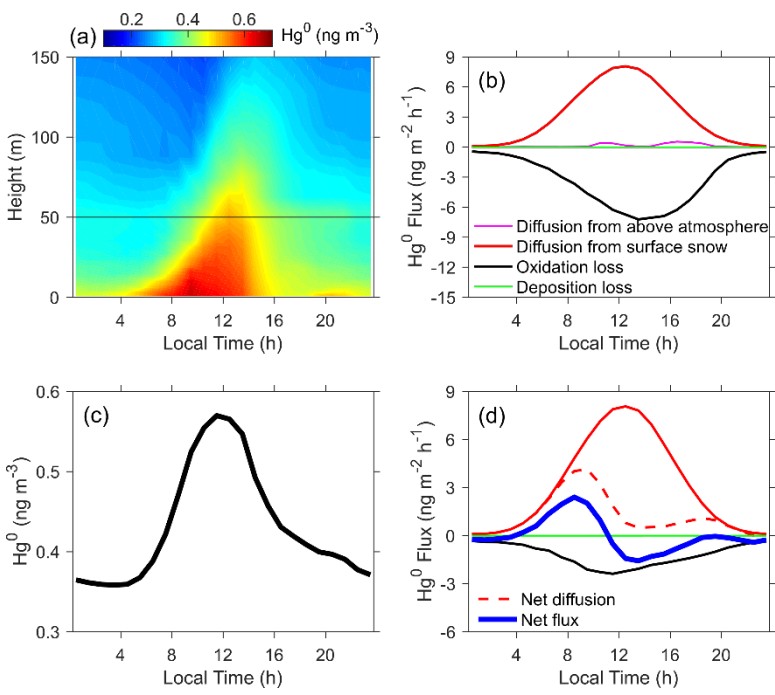

**Figure 5. Summertime diurnal cycles of Hg⁰ concentrations and fluxes.** (a) the modeled vertical distributions of Hg⁰ concentrations in near-surface air, (b) the modeled Hg⁰ fluxes in the inversion/mixed layers, (c) the modeled Hg⁰ concentration averaged for 0–50 m above snow surface, and (d) the modeled Hg⁰ fluxes for the air in 0–50 m above snow surface. Model results from the scenario BR_HH_14d are shown.





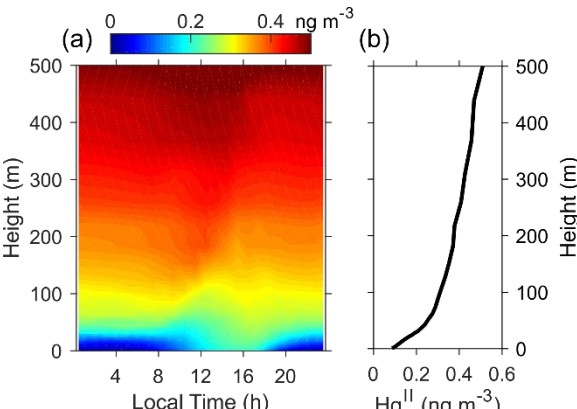

**Figure 6. Summertime diurnal and vertical profiles of atmospheric Hg$^{II}$ concentrations.** (a) shows both diurnal and vertical distributions and (b) only shows the average vertical profile. Model results from the scenario BR_HH_14d are shown.

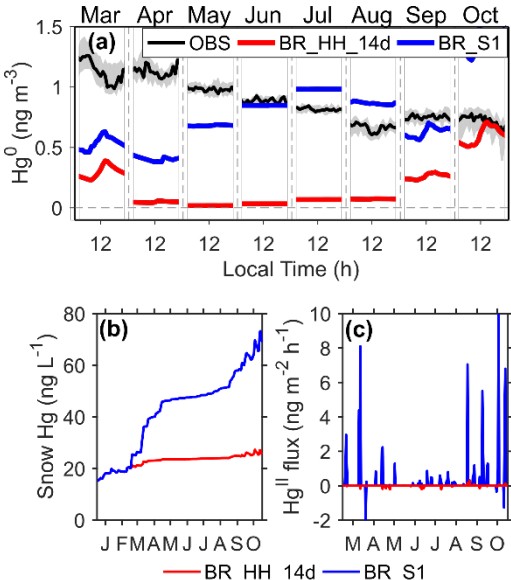

**Figure 7. Possible impacts of warming events on mercury concentrations in the non-summer period.** (a) shows Hg$^0$ observations at 25 cm above snow at Dome C and the shaded regions indicate 25% and 75% percentiles. The modeled Hg$^0$ concentrations from BR_HH_14d and BR_S1 are also shown. (b) shows surface snow mercury concentrations from BR_HH_14d and BR_S1. (c) shows the exchange fluxes of Hg$^{II}$ from the free troposphere modeled by BR_HH_14d and BR_S1.



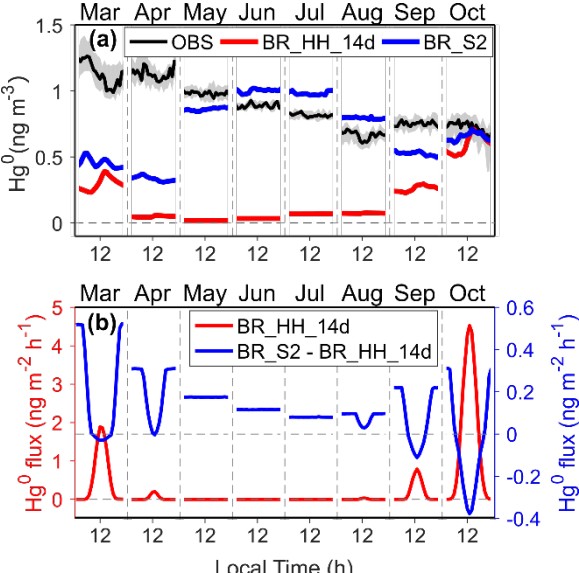

**Figure 8. Possible impacts of snow mercury dark reduction on Hg⁰ concentrations and fluxes in the non-summer period.** (a) shows Hg⁰ observations at 25 cm above snow surface at Dome C, and the shaded regions indicate 25% and 75% percentiles. The modeled Hg⁰ concentrations from BR_HH_14d and BR_S2 are also shown. (b) shows the modeled Hg⁰ snow-to-air diffusion fluxes from BR_HH_14d (left axis), and the difference of snow-to-air diffusive fluxes between BR_S2 and BR_HH_14d (right axis).

**Table 1. Gas phase mercury reactions used in the mercury model.**

| No. | Reaction | Rate constant$^a$ | Reference |
|---|---|---|---|
| *R1* | $Hg^0 + O_3 \rightarrow Hg^{II}$ | $k_1 = 1.7 \times 10^{-18}$ (*upper*) | (Schroeder et al., 1991) |
| | | $k_1 = 3 \times 10^{-20}$ (*lower*) | (Hall, 1995) |
| *R2* | $Hg^0 + OH \rightarrow Hg^{II}$ | $k_2 = 3.2 \times 10^{-13} \times (T/298)^{-3.06}$ (*upper*) | (Goodsite et al., 2004) |
| | | $k_2 = 8.7 \times 10^{-14}$ (*lower*) | (Sommar et al., 2001) |
| *R3* | $Hg^0 + Br \rightarrow Hg^IBr$ | $k_3 = 3.2 \times 10^{-12}$ (*upper*) | (Ariya et al., 2002) |
| | | $k_3 = 1.46 \times 10^{-32} \times (T/298)^{-1.86} \times [M]$ (*lower*) | (Donohoue et al., 2006) |
| *R4$^b$* | $Hg^IBr \rightarrow Hg^0 + Br$ | $k_4$ [s$^{-1}$] = $k_3 / K_{eq}$ | (Dibble et al., 2012) |
| *R5* | $Hg^IBr + Br \rightarrow Hg^0 + Br_2$ | $k_5 = 3.9 \times 10^{-11}$ | (Balabanov et al., 2005) |
| *R6* | $Hg^IBr + NO_2 \rightarrow Hg^{II}$ | $k_6 = 8.6 \times 10^{-11}$ | (Dibble et al., 2012; Wang et al., 2014) |
| *R7* | $Hg^IBr + OH \rightarrow Hg^{II}$ | $k_7 = 6.3 \times 10^{-11}$ | (Dibble et al., 2012; Wang et al., 2014) |
| *R8* | $Hg^IBr + HO_2 \rightarrow Hg^{II}$ | $k_8 = 8.2 \times 10^{-11}$ | (Dibble et al., 2012; Wang et al., 2014) |
| *R9* | $Hg^IBr + Br \rightarrow Hg^{II}$ | $k_9 = 6.3 \times 10^{-11}$ | (Dibble et al., 2012; Wang et al., 2014) |
| *R10* | $Hg^IBr + BrO \rightarrow Hg^{II}$ | $k_{10} = 1.1 \times 10^{-10}$ | (Dibble et al., 2012; Wang et al., 2014) |

$^a$Rate constants are in cm$^3$ molecule$^{-1}$ s$^{-1}$ unless otherwise stated. T represents temperature in K. [M] is the number density of air in molecule cm$^{-3}$. The "*upper*" and "*lower*" indicate the highest and lowest reaction rate constants determined by different kinetic studies (for a review, see Ariya et al. (2015)), respectively. The uncertainty ranges of reaction rate constants of *R4–R10* are unknown as only computational kinetic data are available for these reactions (Jiao and Dibble, 2017). $^b$R3 and R4 are a pair of reversible reactions. $K_{eq}$ (= $9.14 \times 10^{-24}$ $e^{7801/T}$ cm$^3$ molecule$^{-1}$) is the equilibrium constant estimated by Dibble et al. (2012), which is very close to the value of $9.25 \times 10^{-23} \times (T/298)^{-2.76}$ $e^{7292/T}$ cm$^3$ molecule$^{-1}$ calculated by Goodsite et al. (2012).