# Peer review of "Understanding mercury oxidation and air-snow exchange on the East Antarctic Plateau: A modeling study"

_Atmospheric Chemistry and Physics, 2018_

## Referee Comment (RC1) · Anonymous Referee #1 · 22 Aug 2018

Over the last years exceptional measurements of mercury in air and snow have been performed in the arctic. However, current numerical models are not able to reproduce let alone explain the observed annual and diurnal variability in Hg concentrations in this region. The exact processes governing the fate of mercury in Polar Regions are currently not well understood. However, in order to understand and predict global mercury cycling it is necessary to investigate the impact of relevant chemical and physical processes. In this paper by Shaojie et al., the authors employ a box modeling approach to investigate the impact of different processes on mercury cycling in the arctic. The results of this study will benefit both the modeling and measurement community. The paper is clear and concise and overall well written. Thus, I recommend publication of

this manuscript with a few minor comments.

P2 l26-29: (quite technical, but in my opinion an important issue that should be mentioned) You should also mention physical/numerical issues of spherical global domains at the poles. To my knowledge none of the global Hg models has been run with a rotated grid to optimize transport in the area of interest. I guess this is also the reason you are using regional model data for this study. P5 l4: A uniform O3 profile for the whole year? Did you consider stratospheric O3 intrusions or O3 depletion events? P5 l20: Why didn't you use the inorganic bromine measurements to adjust the modelled Br/BrO concentrations fields? I think you should add this as an additional sensitivity run. (See also p12 l11-19) Please give an overview of all model sensitivity run in a separate table. I is not enough to explain that in the fig. 3 capture. Fig 1: What about dark oxidation is that included in the net. dark red rate? Later on you perform a dark oxidation experiment. Still, it would help to mention that the other scenarios do not include any dark oxidation rates. I have the opinion that you should go over your conclusions section once more. The lessons you draw from your study seem a bit too general at times: e.g. "It is also important to reduce uncertainties in existing chemical kinetic parameters of bromine oxidation mechanisms. Finally, expecting your model to be highly performant. Have you thought about a monte-carlo approach for restraining reaction and exchange rates?

---

## Referee Comment (RC2) · Anonymous Referee #2 · 19 Sep 2018

The manuscript "Understanding mercury oxidation and air-snow exchange on the East Antarctic Plateau: A modelling study" by Song et al. deal with box model calculations with the aim to reproduce the diurnal variation of mercury in the atmosphere surrounding the snow pack and in connection with changes in surface snow concentration. The role of the polar area is particularly important for global mercury cycle and, the process occurring in these remote regions, are attracting more attention. The poles have been suggested to be a sink (during winter) and source of mercury during summer. The rapid atmospheric chemical reaction that mercury could undergoes, make this elements particularly difficult to study, and full understand its biogeochemical cycle is not always an easy task. In addition mercury is not stable after deposition in surface snow ad can un-

dergoes to rapid re-emission from snow surface impacting the polar atmosphere. The study presented by Song and co-author is the first attempt to reproduce the diurnal variation of mercury in connection with snow. Thought there are assumptions adopted in the box model calculation the authors success to reproduce the average monthly and diurnal observations at Dome C, for winter time some bias have been suggest might due to the dark mercury reaction. Thus, I recommend publication of this manuscript with few minor comments.

Considering the lack of data for specific atmospheric species, important for the box model calculation (such as BrO), together with the statements made by the authors (for example do not consider the wet depositions), I recommend to include a table with all the assumption made to give a clear view and the limit to a possible reader. In addition this table might be useful for promote additional field measurements helpful for better constrain the model simulation.

Specific comments:

Page 3, line 15. The authors claim that they do not consider the wet deposition in Dome C. I am agree with them since the wet deposition are rare and more often during wintertime. However I would like to ask if the authors have considered the diamond dust deposition. This phenomenon seems quite efficient in removing Hg from the atmosphere and can occur pretty often during summer time.

Page 5, line 13. Why didn't you use the inorganic bromine measurements to adjust the modelled Br/BrO concentrations fields (agree with the anonymous referee #1)

Page 6, line 14. The wind and the snow proprieties are not included in the study but they should play a non-negligible role in the mercury re-emission from the snow pack. For example the thickness of the surface wind packed snow layer could have an impact in gas release as well the wind strength could have a different pumping effect. Data on physical snow proprieties in Dome C exist and should be consider for future mercury model exercise.

Page 9, line 1. Field experiments suggest that the mercury lifetime in surface snow (2-3 cm) might be much less than 16 days.

---

## Author Comment (AC1) · 26 Sep 2018

Shaojie Song on behalf of all the authors

acp-2018-436 "Understanding mercury oxidation and air-snow exchange on the East Antarctic Plateau: A modeling study"

Comments are in black and responses are in blue.

**Response to Anonymous Referee #1**

Over the last years exceptional measurements of mercury in air and snow have been performed in the arctic. However, current numerical models are not able to reproduce let alone explain the observed annual and diurnal variability in Hg concentrations in this region. The exact processes governing the fate of mercury in Polar Regions are currently not well understood. However, in order to understand and predict global mercury cycling it is necessary to investigate the impact of relevant chemical and physical processes. In this paper by Shaojie et al., the authors employ a box modeling approach to investigate the impact of different processes on mercury cycling in the arctic. The results of this study will benefit both the modeling and measurement community. The paper is clear and concise and overall well written. Thus, I recommend publication of this manuscript with a few minor comments.

Thanks for these positive comments for our manuscript. Our responses to specific comments are provided below.

P2 l26-29: (quite technical, but in my opinion an important issue that should be mentioned) You should also mention physical/numerical issues of spherical global domains at the poles. To my knowledge none of the global Hg models has been run with a rotated grid to optimize transport in the area of interest. I guess this is also the reason you are using regional model data for this study.

This is a good point. One of the reasons that we use meteorological output from MAR is that MAR is a polar-oriented atmospheric model, with a much more detailed representation of the stable boundary layer than that of a global model. We have added it in P2 l26-29: "*Overall, these observed seasonal and diurnal features of atmospheric mercury on the plateau are not well understood and not reproduced by global chemical transport models, likely due to their imperfect representations of boundary layer dynamics and chemical reaction pathways (Angot et al., 2016a) and to the singularity of their longitude–latitude grid at the poles.*"

P5 l4: A uniform $O_3$ profile for the whole year? Did you consider stratospheric $O_3$ intrusions or $O_3$ depletion events?

The mercury model specifies the temporal variation of $O_3$ based on in situ measurements conducted at Dome C, and therefore the influence of stratospheric intrusions and local depletions is considered. To make this clear, we revise this sentence to (see P5 l3-6): "*The temporal variations of $O_3$ and $NO_x$ are specified based on in situ measurements in near-surface air (Angot et al., 2016c; Legrand et al., 2016a; Helmig et al., 2018), and a uniform $O_3$ vertical profile within the inversion/mixed layers is assumed, consistent with aircraft observations on the plateau (Slusher et al., 2010; Legrand et al., 2016a).*"

P5 l20: Why didn't you use the inorganic bromine measurements to adjust the modelled Br/BrO concentrations fields? I think you should add this as an additional sensitivity run. (See also p12 l11-19)

A quantitative adjustment of BrO (and the resulting Br) concentration fields using the modeled and measured (by Legrand et al. 2016 JGR) total inorganic bromine ($Br_y$) concentrations is difficult mainly due to two factors: (1) The inconsistency in bromine species. The $p$-TOMCAT modeled $Br_y$ refers to the sum of Br, HBr, BrO, HOBr, $Br_2$, $BrNO_2$, and $BrONO_2$, whereas the measured total inorganic bromine trapped by mist chambers and denuder tubes may refer to $Br_y$ or $Br_y^*$ ($[Br_y^*]$ $\approx [Br_y] – 1.1[Br_2] – 0.6[BrO]$); and (2) It is unclear whether and how much $BrNO_2$ and $BrONO_2$ contribute to the discrepancy of total inorganic bromine between the measurements and $p$-TOMCAT model.

Therefore, we only include a sensitivity simulation in order to qualitatively evaluate this potential bias in the mercury model. We have made this clearer in Section 3.4 (see P12 I17-19): "*In order to qualitatively evaluate this potential bias in BrO (and Br) concentrations, we have conducted a sensitivity simulation that reduces BrO (and thus Br) concentrations in fall by a factor of 3. We find that reducing BrO in fall could increase the modeled air Hg$^0$ concentrations during the fall and winter months (Fig. S15 in the Supplement).*"

Please give an overview of all model sensitivity run in a separate table. I is not enough to explain that in the fig. 3 capture.

We provided an overview of the modeling scenarios as a separate table in the supplement. We have made this clearer in P8 I2-3: "*In total, we ran 24 model sensitivity scenarios (Table S1 in the Supplement).*"

Fig 1: What about dark oxidation is that included in the net. dark red rate? Later on you perform a dark oxidation experiment. Still, it would help to mention that the other scenarios do not include any dark oxidation rates.

We only include reduction (either photolytic- or dark-) of snow mercury in the model, mainly because production of Hg$^0$ is required to sustain atmospheric Hg$^0$ levels. It can be regarded as a net reaction rate if any snow mercury oxidation process occurs in the real world. The dark reduction of surface snow Hg$^{II}$ may be only important for the non-summer period (Sect. 2.5), and we have made this clearer in the caption of Fig. 1.

I have the opinion that you should go over your conclusions section once more. The lessons you draw from your study seem a bit too general at times: e.g. "It is also important to reduce uncertainties in existing chemical kinetic parameters of bromine oxidation mechanisms.

We have revised this section and made our suggestions for further research clearer (P13 I9-16): "*In order to obtain a better understanding of mercury cycling over the East Antarctic plateau, we suggest several areas for future research. (1) It is essential to better constrain the concentration levels of bromine species, especially BrO$_x$, through more field experiments and modeling studies. (2) It is important to reduce uncertainties in existing chemical kinetic parameters of bromine oxidation mechanisms. The rate constant of Hg$^0$ reaction with Br from existing theoretical and experimental studies varies by a factor of 4. (3) Our modeling indicates relatively high*

*atmospheric Hg$^{II}$ concentrations in summer, which remains to be verified by additional field measurements. (4) A better characterization of atmospheric vertical transport during the non-summer period is needed, in particular the role of intermittent warming events. (5) The chemical mechanisms and reaction rates for snow mercury processes, including photo- and dark-reduction, should be further investigated*".

Finally, expecting your model to be highly performant. Have you thought about a monte-carlo approach for restraining reaction and exchange rates?

We considered a Monte Carlo approach, but decided to use a simpler sensitivity test approach. This is mainly because the probability distributions of some important physical and chemical processes/parameters, for example the vertical turbulent diffusivity during the warming events, are difficult to obtain. We may be able to apply a Monte Carlo approach in the future when a better understanding of the physiochemical mercury processes becomes available.

**Response to Anonymous Referee #2**

The manuscript "Understanding mercury oxidation and air-snow exchange on the East Antarctic Plateau: A modelling study" by Song et al. deal with box model calculations with the aim to reproduce the diurnal variation of mercury in the atmosphere surrounding the snow pack and in connection with changes in surface snow concentration. The role of the polar area is particularly important for global mercury cycle and, the process occurring in these remote regions, are attracting more attention. The poles have been suggested to be a sink (during winter) and source of mercury during summer. The rapid atmospheric chemical reaction that mercury could undergoes, make this elements particularly difficult to study, and full understand its biogeochemical cycle is not always an easy task. In addition mercury is not stable after deposition in surface snow ad can undergoes to rapid re-emission from snow surface impacting the polar atmosphere. The study presented by Song and co-author is the first attempt to reproduce the diurnal variation of mercury in connection with snow. Thought there are assumptions adopted in the box model calculation the authors success to reproduce the average monthly and diurnal observations at Dome C, for winter time some bias have been suggest might due to the dark mercury reaction. Thus, I recommend publication of this manuscript with few minor comments.

Thanks for these positive comments for our manuscript. Our responses to specific comments are provided below.

Considering the lack of data for specific atmospheric species, important for the box model calculation (such as BrO), together with the statements made by the authors (for example do not consider the wet depositions), I recommend to include a table with all the assumption made to give a clear view and the limit to a possible reader. In addition this table might be useful for promote additional field measurements helpful for better constrain the model simulation.

This is a very good suggestion. We have added such a table summarizing the assumptions and simplifications made in the mercury model. It is Table 1 in the revised manuscript.

**Table 1. Major assumptions and simplifications made in the mercury model.**

| Description | Note |
| --- | --- |
| *Physical or chemical processes not considered* | |
| Horizontal transport | The model is not expected to capture day-to-day variability |
| Photoreduction of $Hg^{II}$ in aqueous cloud/aerosol | The air is cold and dry |
| Wet deposition of $Hg^{II}$ | Large uncertainty in its parameterization |
| Exchange with deep snowpack Hg | The diffusive transfer is expected to be slower |
| *Simplifications for specific species or parameters* | |
| Free tropospheric Hg concentration | Specified based on CTMs |
| $HO_x$ concentration | Estimated based on OPALE measurements, NO, and $J(NO_2)$ |
| $BrO_x$ concentration | Specified based on CTMs |
| Air turbulent diffusion coefficient ($K_z$) | Modeled by MAR (with an optional adjustment for warming events) |
| Dry deposition velocities ($V_d$) | Typical values from the literature |
| Depth of surface snow layer | Specified based on $e$-folding light penetration depth |
| Air–snow molecular diffusion coefficient ($D_m$) | Typical value from the literature |
| Air–snow turbulent diffusion coefficient ($D_t$) | Parameterized based on surface level turbulent kinetic energy (TKE) |

Specific comments:

Page 3, line 15. The authors claim that they do not consider the wet deposition in Dome C. I am agree with them since the wet deposition are rare and more often during wintertime. However I would like to ask if the authors have considered the diamond dust deposition. This phenomenon seems quite efficient in removing Hg from the atmosphere and can occur pretty often during summer time.

We agree that snowfall and diamond dust deposition events may be an efficient pathway for mercury deposition given the recent study by Spolaor et al. (2018). This process (and the fate of deposited mercury) is still uncertain and also difficult to parameterize in the model, and is not included in the current study. We have made this clear in the manuscript (P3 I15-19): "*Wet deposition is not considered due to low snow accumulation rates and large uncertainty in parameterizing this process (France et al., 2011; Palerme et al., 2017). Note that Spolaor et al. (2018) have recently suggested that frequent snowfall and diamond dust (tiny ice crystals) events in summer may lead to quick mercury deposition. However, a quantitative parameterization for this process has not been available, and it is thus not included in this model*".

Page 5, line 13. Why didn't you use the inorganic bromine measurements to adjust the modelled Br/BrO concentrations fields (agree with the anonymous referee #1)

A quantitative adjustment of BrO (and the resulting Br) concentration fields using the modeled and measured (by Legrand et al. 2016 JGR) total inorganic bromine ($Br_y$) concentrations is difficult mainly due to two factors: (1) The inconsistency in bromine species. The *p*-TOMCAT modeled $Br_y$ refers to the sum of Br, HBr, BrO, HOBr, $Br_2$, $BrNO_2$, and $BrONO_2$, whereas the measured total inorganic bromine trapped by mist chambers and denuder tubes may refer to $Br_y$ or $Br_y^*$ ($[Br_y^*]$ $\approx [Br_y] - 1.1[Br_2] - 0.6[BrO])$; and (2) It is unclear whether and how much $BrNO_2$ and $BrONO_2$ contribute to the discrepancy of total inorganic bromine between the measurements and *p*-TOMCAT model.

Therefore, we only include a sensitivity simulation in order to qualitatively evaluate this potential bias in the mercury model. We have made this clearer in Section 3.4 (see P12 I17-19): "*In order to qualitatively evaluate this potential bias in BrO (and Br) concentrations, we have conducted a sensitivity simulation that reduces BrO (and thus Br) concentrations in fall by a factor of 3. We find that reducing BrO in fall could increase the modeled air Hg^0 concentrations during the fall and winter months (Fig. S15 in the Supplement).*"

Page 6, line 14. The wind and the snow proprieties are not included in the study but they should play a non-negligible role in the mercury re-emission from the snow pack. For example the thickness of the surface wind packed snow layer could have an impact in gas release as well the wind strength could have a different pumping effect. Data on physical snow proprieties in Dome C exist and should be consider for future mercury model exercise.

We agree that wind and snow properties play a non-negligible role in the air-snow mercury exchange, and that a more explicit consideration of these properties in the model may be important. We find several parameters in the estimation of vertical wind pumping, such as the height and wavelength of sastrugi and the permeability of surface snowpack, are uncertain and may be subject

to some currently unknown temporal variability. Thus, we use in the current model a more simple approach from Durnford et al. (2012), based on the turbulent kinetic energy. This approach may have considered the influence of surface wind properties (partially and implicitly) but not snow properties. Following your suggestion, we have made a recommendation for a more explicit consideration of air and snow properties' effects in P6 I28-29: "*A more explicit consideration of the influence of air and snow properties on air-snow exchange is recommended for future mercury modeling studies.*"

Page 9, line 1. Field experiments suggest that the mercury lifetime in surface snow (2-3 cm) might be much less than 16 days.

We agree that the lifetime of snow mercury in the top 2-3 cm can be much less when compared with that for the top 20 cm (assumed in this study based on the *e*-folding depth of solar radiation penetration). The mercury lifetime of 16 days at South Pole was estimated according to a surface layer of 15 cm in Brooks et al. (2008), which agreed well with the assumption for this study. We have made this clearer in P8 I34-P9 I2: "*The photoreduction rates of surface snow (top 20 cm) $Hg^{II}$ in BR_HH_14d ($\tau_{PR}$ of 2 weeks) agree well with observations at South Pole in Brooks et al. (2008), who estimated a lifetime of surface snow mercury (assumed to be the top 15 cm) of ~16 days.*"